# Orthogonal programming of heterogeneous micro-mechano-environments and geometries in three-dimensional bio-stereolithography

Hang Yin[1], Yonghui Ding[1], Yao Zhai[1], Wei Tan[1] & Xiaobo Yin[1,2]

Engineering heterogeneous micro-mechano-microenvironments of extracellular matrix is of great interest in tissue engineering, but spatial control over mechanical heterogeneity in three dimensions is still challenging given the fact that geometry and stiffness are inherently intertwined in fabrication. Here, we develop a layer-by-layer three-dimensional (3D) printing paradigm which achieves orthogonal control of stiffness and geometry by capitalizing on the conventionally adverse effect of oxygen inhibition on free-radical polymerization. Controlled oxygen permeation and inhibition result in photo-cured hydrogel layers with thicknesses only weakly dependent to the ultraviolet exposure dosage. The dosage is instead leveraged to program the crosslink density and stiffness of the cured structures. The programmable stiffness spans nearly an order of magnitude ($E \sim$ 2–15 kPa) within the physiologically relevant range. We further demonstrate that extracellular matrices with programmed micro-mechano-environments can dictate 3D cellular organization, enabling in vitro tissue reconstruction.

[1] Department of Mechanical Engineering, University of Colorado, Boulder, CO 80309, USA. [2] Materials Science and Engineering Program, University of Colorado, Boulder, CO 80309, USA. These authors contributed equally: Hang Yin, Yonghui Ding. Correspondence and requests for materials should be addressed to W.T. (email: wtan@colorado.edu) or to X.Y. (email: xiaobo.yin@colorado.edu)

Precise spatial organization of tissue mechanics creates heterogeneous micro-mechano-environments optimal for diverse functions such as morphogenesis[1] and regeneration[2]. Mechanical heterogeneity is also closely associated with development of cardiovascular disease[3], and breast tumorigenesis[4]. Engineering an extracellular microenvironment that provides the level of mechanical, structural, and biochemical heterogeneity found in native tissues is of great interest for tissue and organ replacement, drug screening, and disease modeling. Photo-liable reactions have been widely exploited to spatially soften or stiffen hydrogel structures but have been limited to two-dimensional (2D) geometries and multi-stage polymerization processes[5–8]. Before optical saturation, geometry and stiffness are, however, inherently intertwined in three-dimensional (3D) photopolymerization because of the different extents of light scattering and outgrowth of photopolymerization at different levels of exposures[9], making it fundamentally challenging to prepare complex 3D structures with independently defined heterogeneous micro-mechano-environments and geometries[10–14].

Oxygen inhibition had been regarded as a major hurdle to free-radical polymerization (FRP) because it causes incomplete curing and tacky surfaces[15]. In contrast, well-controlled oxygen inhibition has recently shown great potential for high-throughput photo-polymerizations[16]. Continuous liquid interface production (CLIP) has been recently demonstrated by leveraging an inhibition layer introduced on top of an oxygen-permeable window[17]. Utilizing a similar concept, a high-resolution micro-CLIP has enabled 3D printing of bioresorbable vascular devices with microscale resolution[18,19]. Controlled oxygen permeation can also be an asset for engineering mechanical properties in multi-stage photo-polymerizations[20].

Here, we develop a layer-by-layer 3D printing paradigm where we purposely introduce an oxygen inhibition layer between a cured polymer structure and an oxygen-permeable window to physically limit the curing thickness during the layer-by-layer construction process in stereolithography. The thickness of the cured layer under controlled oxygen inhibition becomes nearly insensitive to the exposure dosage, which instead modulates the local crosslink density and, therefore, stiffness. This new technique allows us to print 3D structures with orthogonally patterned geometry and stiffness. To demonstrate the utility of this system, the vascular smooth muscle cells are seeded on 3D-printed extracellular matrix to investigate how the programmed micro-mechano-environments dictate 3D cellular organization and in vitro tissue reconstruction.

## Results

### The oxygen inhibition-assisted 3D printing paradigm.
To demonstrate the oxygen inhibition enabled stiffness control, we use layer-by-layer stereolithography, shown schematically in Fig. 1a. Continuous and layerless stereolithography can also be similarly implemented for a higher throughput[17,21]. A 3D model of a complex object is first sliced into multiple layers of even thicknesses. These layers are then sequentially printed by projecting corresponding gray-scale images into the hydrogel precursor solution with a dynamic micro-mirror device (DMD). The curing zone between the oxygen inhibition layer and the cured region is physically limited (Fig. 1b), resulting in uniform curing thickness across each individual layer when the UV exposure dosage is above the threshold. The gray-scale intensity of the projected image is then leveraged to dictate spatial crosslink density and stiffness. A motorized stage is used to pull the cured structure out of the image plane where UV light is focused. Complex 3D models can, therefore, be printed efficiently. Figure 1a inset shows an example of an arbitrary 3D structure, which

was printed within 10 min. Poly(ethylene glycol)dimethacrylate (PEGDMA, $M_w$ 750) was used as an exemplary free-radical, UV-curable hydrogel material, which is broadly used in biomedical applications for its excellent biocompatibility, antifouling nature, and well-controlled mechanical properties[22]. The photoinitiator, lithium phenyl-2,4,6-trimethylbenzoylphosphinate (LAP), was also used for its high water solubility and exceptional photo-initiating efficiency, essential for fast polymerization and continuous printing[23].

A 10-μm-thick oxygen-permeable polydimethylsiloxane (PDMS) film was coated on a glass substrate underneath the hydrogel precursor solution (Fig. 1b). Oxygen from surrounding air freely diffuses through the porous walls of PDMS and inhibits FRP of PEGDMA by reacting with radical species to form chain-terminating peroxide molecules[15]. The double bond conversion rate of PEGDMA is a function of both the UV irradiation dosage and the depth into the solution, given a constant oxygen concentration of $0.35 \text{ mol m}^{-3}$ in PDMS film and oxygen diffusion rate of $2.84 \times 10^{-11} \text{ m}^2 \text{ s}^{-1}$ in crosslinked PEGDMA[24]. As shown in Fig. 1c, when exposure dosage is below a threshold $\sim 20 \text{ mJ cm}^{-2}$, the presence of oxygen prevents the FRP and no double bond conversion is observed. An oxygen inhibition layer is observed on top of the PDMS at high exposure dosages. The thickness of the inhibition layer is reduced slightly with increasing dosages; however, it is nearly constant when dosage is high. In stark contrast to the weakly changing thickness, the double bond conversion rate rapidly increases with exposure dosages in the curing zone. The drastically different behaviors of the curing thickness and the double bond conversion rate in response to increasing exposure dosage enable unique orthogonal control over the 3D geometry and stiffness in oxygen inhibition assisted stereolithography. Projection of "grayscale" intensity maps of the sliced images results in spatially varying stiffness in the cured structures, while the threshold defines the boundary and geometry of the printed structures.

To demonstrate that the 3D printing method can control stiffness and geometry independently due to the oxygen inhibition layer, a buffalo logo was printed (Fig. 1d) with a binary exposure scheme. Features of the logo were exposed with a UV dosage of $72 \text{ mJ cm}^{-2}$ while regions in the background were exposed with $44 \text{ mJ cm}^{-2}$. Five identical exposures were used to form a uniform thickness of $\sim 350 \text{ μm}$. Variation of feature height between the two regions is $< 1\%$ (as measured by profilometer in Fig. 1e), suggesting even thickness control during the printing. However, sharp optical contrast is observed (Fig. 1d), indicating a strong difference in crosslink density[6]. The difference in optical contrast is further quantified by a line profile of intensity in the bright-field image (Fig. 1e). Altogether, we demonstrate that the binary dosage scheme results in uncorrelated patterning of stiffness and geometry (cured thickness).

### Quantitative analysis of programmed stiffness and geometry.
The mechanical properties of cured polymer are highly sensitive to the crosslink density. We measured the stiffness (Young's modulus $E$) and the surface morphology of the printed structures in phosphate-buffered saline (PBS) solution using atomic force microscopy (AFM). All stiffness measurements were performed after overnight ($> 12 \text{ h}$) swelling post-printing to ensure that the polymerization of remaining free radicals as well as hydrogel swelling were complete. Multiple binary exposures (a total thickness of $\sim 500 \text{ μm}$) were used to ensure the accuracy of Hertz-model-based stiffness measurement by AFM[25]. Line features (width: 10 μm; length: 1 mm) were exposed with UV dosage of $72 \text{ mJ cm}^{-2}$ while regions in the background are exposed with $44 \text{ mJ cm}^{-2}$. Notably, the sharp optical contrast in bright-field microscope image

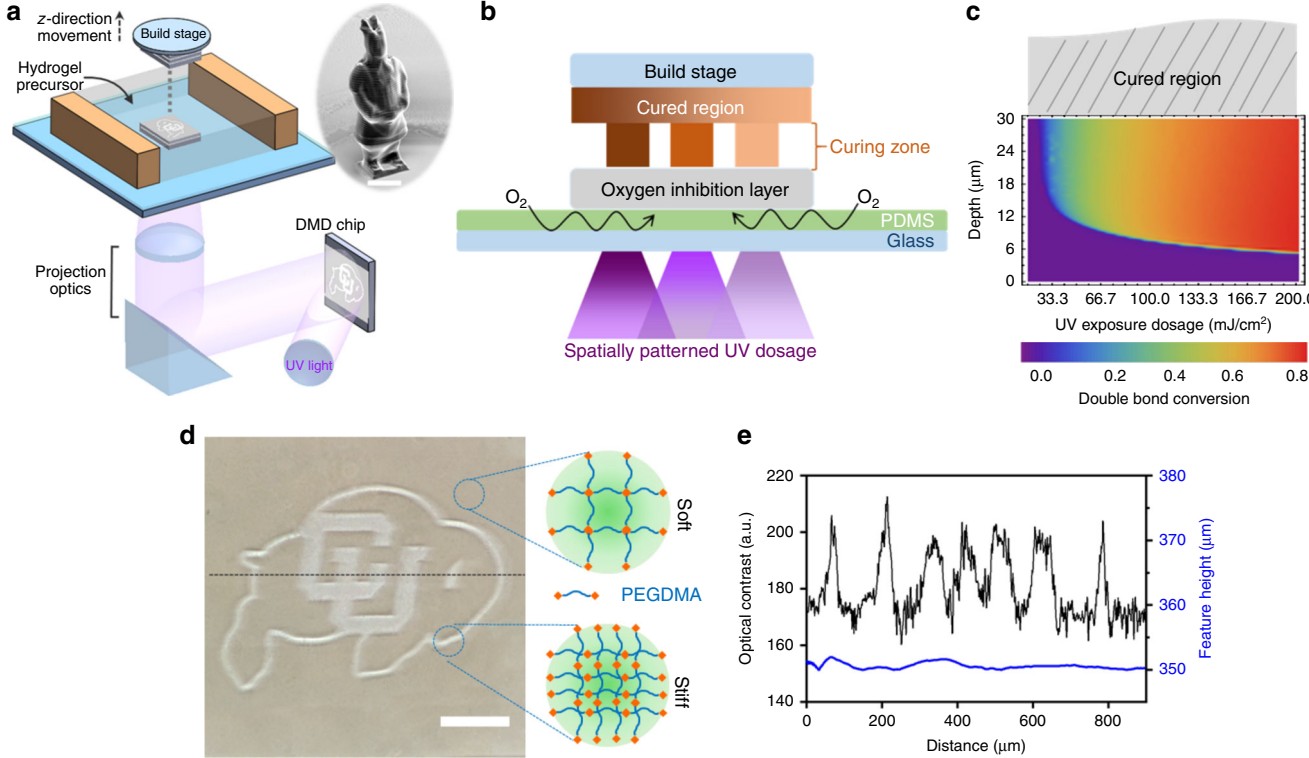

**Fig. 1** Orthogonal programming of matrix stiffness and geometry via oxygen inhibition-assisted stereolithography. **a** Schematic set-up of digital projection stereolithographic 3D printing system where hydrogel precursor solution is cured layer-by-layer through UV exposure. Inset is a SEM image of a 3D-printed complex object. Scale bar is 500 μm. **b** Schematic of oxygen inhibition-assisted printing, in which the curing zone is physically limited between the cured region and the oxygen inhibition layer. **c** Depth profile of double bond conversion rate under different UV exposure dosages. The thickness of oxygen inhibition layer is weakly dependent to the exposure dosages, and so does the curing thickness. The double bond conversion rate rapidly increases with the dosage when dosage is above the threshold. **d** Bright-field optical image of a printed buffalo logo with independently patterned stiffness and geometry (binary stiffness but flat surface). High optical contrast indicates the strong differences in crosslink density and, therefore, the stiffness. Scale bar is 200 μm. **e** Quantification of optical contrast (black line) and geometry (blue line) variation along the dotted line in **b** reveals sharp differences in contrast (stiffness) but little feature height variation ( <1%)

(Fig. 2a) correlates well with stiffness patterns observed in AFM stiffness mapping, where $E \sim 11$ kPa for line features and $E \sim 5$ kPa for background regions (Fig. 2b). In contrast, limited topographic variation ( <600 nm) between the stiff lines and soft background regions is revealed by AFM topography mapping (Fig. 2c). The results affirm that the stiffness and geometry can be independently controlled with our approach. A range of 0.1–17 kPa matrix stiffness encompasses the stiffness of most physiological soft tissues[26]. In Fig. 2d, we show the stiffness and feature height of crosslinked PEGDMA as function of the exposure dosage. The stiffness can be tuned over an order of magnitude ($E \sim 2$–15 kPa) within a physiological-relevant range (solid, black circles in Fig. 2d). The stiffness can be expanded to a wider range using a higher power UV source; however, the detrimental effects of higher levels of UV dosage and free radicals on living cells need to be taken into account when printing hydrogel structures for living cell encapsulation[27]. In contrast, the geometry variation is limited to within 2% for a ~500 μm high structure (open, blue circles in Fig. 2d). With this limited difference in feature height, we observe the trend that stiffer regions have a slightly larger feature height as the results of mitigated oxygen inhibition (reduced oxygen inhibition layer thickness) under stronger UV exposures (Fig. 1c)[17,28]. Further examination of the crosslink density in polymerized PEGDMA hydrogels by swelling ratio measurement and Flory-Rehner theory[29] demonstrates the increasing effective crosslink density with increasing UV dosages (Supplementary Methods and Supplementary Fig. 1), which is in consistent with the AFM stiffness measurement. Moreover,

oxygen diffusion and inhibition also affect the minimal feature sizes that could be achieved[30]. A higher dosage exposure not only results in a stiffer structure but also allows printing a smaller feature. We have achieved approximately 20 μm well-defined rod structures at a dosage of 84 mJ cm$^{-2}$ (Supplementary Fig. 2).

**Programming stiffness and geometry in 3D-printed overhang structures**. Figure 3 shows several simple 3D-printed structures demonstrating orthogonal control over the geometry and stiffness. In Fig. 3a–c, each structure consists of two supporting rods holding a top beam. The supporting rods were printed with the same geometry (diameter: 80 μm; height: 900 μm) but programmed into three different stiffness combinations: stiff/stiff, soft/stiff, and soft/soft. There was no visual difference among printed structures when first observed in PBS solution and ethanol due to buoyancy. As shown in Fig. 3a–c, however, only the stiff rods were strong enough to avoid collapsing when removed from solution and dried in air, while the soft rods collapsed. The beam leaned toward the soft rod in the soft/stiff structure (Fig. 3b) and fell over in the soft/soft structure (Fig. 3c). We further demonstrate the orthogonal control of geometry and stiffness in an exemplary 3D structure, a warrior model with a stiff body but a soft heart inside (Fig. 3d–f). SEM image reveals well-defined topography (Fig. 3e) while the apparent optical contrast in dark-field image reflects the differences between the stiff body (bright) and soft heart (dark) (Fig. 3f). In addition, to show our capability of programming multiple stiffness domains

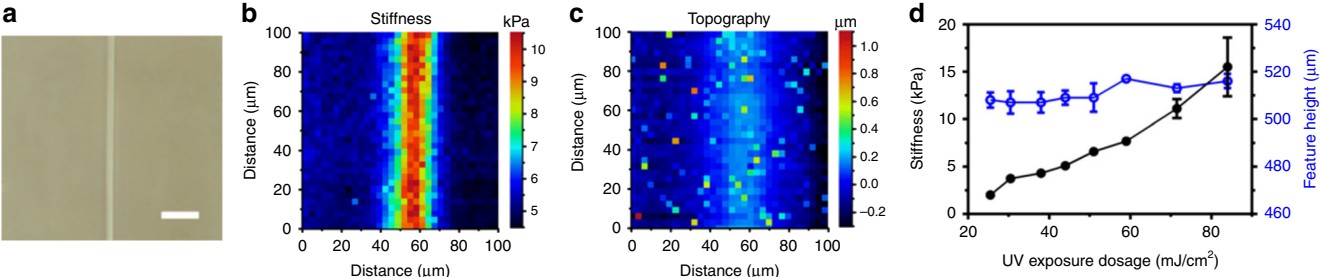

**Fig. 2** Programmed stiffness of photo-curable hydrogels. **a** Bright-field optical image of a stiff line feature (UV exposure dosage of 72 mJ cm$^{-2}$) in a soft hydrogel background (UV exposure dosage of 44 mJ cm$^{-2}$). Scale bar is 100 μm. The total thickness is ~500 μm to improve the accuracy of Hertz-model-based AFM stiffness measurement. Matrix stiffness (**b**) and surface topography (**c**) mappings via AFM measurement reveal sharp contrast in stiffness (stiff line: $E$ ~ 11 kPa; soft background: $E$ ~ 5 kPa) but small topography variations (< 600 nm) for a total thickness of ~500 μm. An AFM cantilever with a 10 μm polystyrene sphere attached and a pre-calibrated stiffness of 0.09 N/m was used for the stiffness results reported here. The force distance curves were recorded and the Hertz-model was used to calculate the stiffness of the polymerized matrix. **d** Quantification of stiffness and geometry variation shows that stiffness (solid black circles) increases from 2–15 kPa with the increase of UV exposure dosage, while the feature height variation (open blue circles) is limited to 2% over UV exposure dosages from 25.5–84 mJ cm$^{-2}$. Error bars represent standard deviation (s.d.); $n = 5$ samples per condition

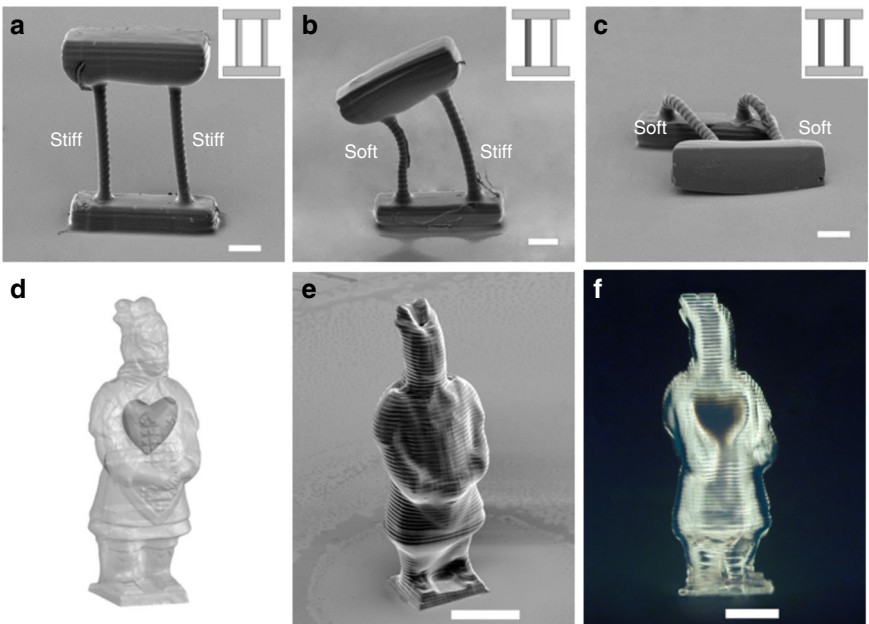

**Fig. 3** Orthogonally programmed stiffness and geometry in 3D-printed overhang structures. A beam was hold by two supporting rods, which were printed with the same geometry (diameter: 80 μm; height: 0.9 mm) but three different stiffness combinations, i.e. **a** stiff/stiff, **b** soft/stiff, and **c** soft/soft (stiff with UV dosage of 72 mJ cm$^{-2}$; soft with UV dosage of 44 mJ cm$^{-2}$). There was no visual difference among printed structures when first observed in PBS solution and ethanol due to buoyancy. When removed from solution and dried in air, the printed soft supporting rods collapsed. Only the stiff/stiff rod structure was strong enough to support the top beam without obvious bending and survived the drying process. Insets are schematic illustrations for functional support structures (light: stiff rod; dark: soft rod). **d**, **e**, and **f** show a printed warrior model with stiff body (UV dosage of 72 mJ cm$^{-2}$) while a soft heart inside (UV dosage of 44 mJ cm$^{-2}$). **d** A CAD model (light: stiff body; dark: soft heart) was modified from an open-source 3D object. **e** SEM image of printed warrior model. **f** Dark-field image shows apparent optical contrast between stiff body and soft heart. Scale bars are **a**, **b**, **c** 200 μm; **e**, **f** 500 μm

simultaneously, four rods with similar geometries were printed with spatially varied UV exposure dosages and, therefore, different stiffness (Supplementary Fig. 3). When released from solution and dried in air, different degrees of bending were observed as the results of spatially varying stiffness. It is also worth noting that no print-through effects[31] were observed in the 3D-printed structures, which benefited from fine control of curing depth enabled by addition of 0.15% UV absorber TINUVIN 234 in the hydrogel precursor solution[17].

**Cellular organization depends on mechanical heterogeneity.** A well-defined synthetic extracellular matrix with spatial control of

matrix mechanical properties may regulate cellular organization, which is critical for reconstituting functional tissues for in vitro modeling as well as implantation for tissue engineering[32,33]. The bovine pulmonary artery smooth muscle cells (bPASMCs), typically involved in the reconstitution of normal or hypertensive diseased vascular tissues[34], are used here as an exemplary system. We first examine the cellular attachment and morphology influenced by programed 2D micro-mechano-environments (Fig. 4a–e). Alternating soft (crosslinked at 44 mJ cm$^{-2}$ with $E$ ~ 5 kPa) and stiff (crosslinked at 72 mJ cm$^{-2}$ with $E$ ~ 11 kPa) strips of PEGDMA hydrogel structures with a line width ~100 μm (Fig. 4a) were printed with the aforementioned approach and the

surfaces were functionalized with the extracellular matrix protein fibronectin. The bPASMCs were then seeded and cultured for 1 day prior to evaluation. Fluorescent images (Fig. 4b) clearly show the preferential attachment of muscle cells on stiff regions, and the quantitative analysis of cell density over a broader area confirms this observation (Fig. 4c). In addition, the attached cells were considerably more elongated with higher aspect ratios (Fig. 4d) and favorably aligned along the strips (Fig. 4e). The effects on cell elongation and alignment are apparent when the strip width is narrow and comparable to cell size while they disappear for a large strip width (e.g. 200 μm). These findings suggest that patterned stiffness provides a strong directional cue to control cellular attachment and morphology on hydrogel surfaces, which is in agreement with the recent observation of selective stem cell[7], and endothelial cell[8] attachment on stiffness controlled surfaces.

The move towards well-defined synthetic 3D systems, however, requires precise control of cell migrations in a more physiological 3D environments[35] and is far more challenging with conventional approaches. The 3D spatial heterogeneous micro-mechano-environment may not only impact cell attachment but also lead to preferential cell migrations in response to the different stiffness profiles. Enabled by our unique oxygen inhibition assisted printing process, Fig. 4f shows schematics of two simple 3D vascular-tube-like structures, which allow demonstration of preferred cell migrations along the stiff 3D structures. The entire structure shown in the upper panel of Fig. 4g is uniformly stiff, which is printed with a dosage of 72 mJ cm$^{-2}$ and an expected modulus of $E \sim 11$ kPa. The control, shown in the lower panel of Fig. 4g is half soft and half stiff with the soft region printed with a dosage of 44 mJ cm$^{-2}$ and an expected modulus of $E \sim 5$ kPa. The bPASMCs were seeded at a high density, and the cells initially

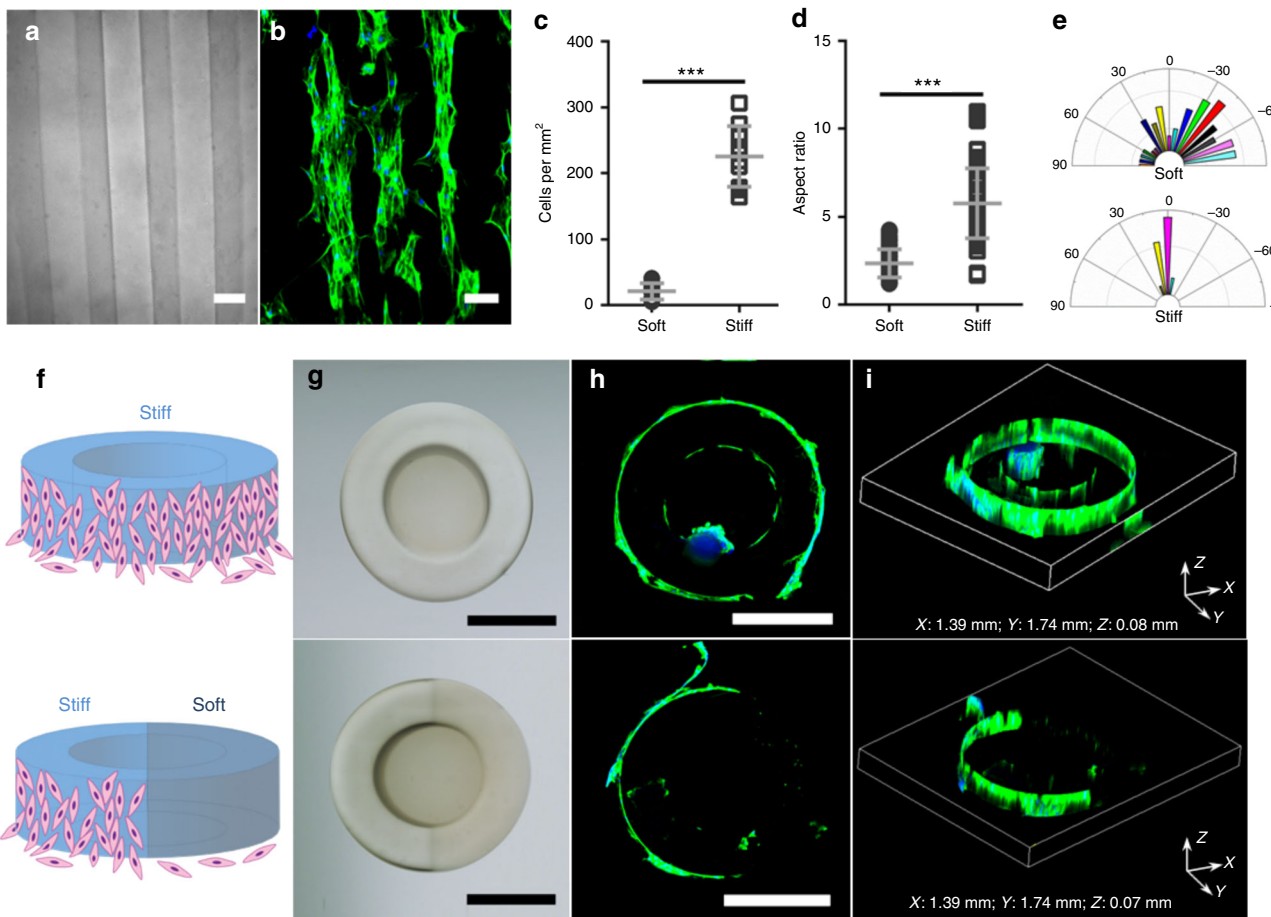

**Fig. 4** Spatial control of cellular organization on structures with independently programmed stiffness and geometry. **a** Bright-field optical image of an alternating 2D soft/stiff stripe patterns (dark line: soft, crosslinked at 44 mJ cm$^{-2}$ with $E \sim 5$ kPa; bright line: stiff, crosslinked at 72 mJ cm$^{-2}$ with $E \sim 11$ kPa; line width: 100 μm). **b** bPASMCs cultured on stiffness-patterned structures prefer to attach to stiff regions vs. soft regions. Quantification of number of cells per mm$^2$ (**c**), aspect ratio (**d**), and alignment (0 degree being perfectly aligned to the pattern) (**e**) showed the majority of cells attached to stiff regions and those cells were more elongated and aligned along the direction of patterned stiffness strips compared to the limited number of cells attached to soft regions. **f** The preferential cell attachment to 2D stiff strip patterns motivates further investigation of 3D cellular organizations. Our hypothesis of directed 3D cell migration by patterned stiffness profiles in printed vascular-tube-like structures. **g** Bright-field microscope images of printed tube structures with either uniformly stiff (upper panel) or soft-stiff (half-half; lower panel) walls. The patterned stiffness is expected to be $E \sim 11$ kPa for stiff walls and $E \sim 5$ kPa for soft walls. The printed tube structures have an outer diameter of 1 mm, an inner diameter of 0.6 mm, and height of 0.1 mm. Confocal X-Y projection (**h**) and 3D view (**i**) showed bPASMCs cultured on printed tube structures preferred to migrate up stiff walls vs. soft walls over the course of the 3-day culture, producing a either 3D whole or half vascular-tube-like tissue structure. Green: f-actin. Blue: nuclei. Scale bars are: **a**, **b** 100 μm; **g**, **h** 500 μm. Error bars represent standard deviation (s.d.); $n = 10$ random images for counting cell number per mm$^2$, and $n > 90$ cells for calculating aspect ratio and alignment. ***$p < 0.001$

formed a layer at the base of the tubes as well as the substrates. Over the course of the 3-day culture, the cells migrated up along the walls of the tubes. As shown in the upper panels of Fig. 4h, i, the bPASMCs migrated up along the wall uniformly and formed 3D cell layers surrounding both the outer and inner walls of the stiff tube, creating a 3D vascular-tube-like tissue structure. Intriguingly, for the control of soft-stiff tube, cells almost exclusively migrated up the stiff regions of the wall, producing a 3D half vascular-tube-like tissue structure (lower panels in Fig 4h, i). The controlled 3D stiffness profiles clearly allow precision in directing cell migration and subsequent cellular organization. Although many other factors including growth-factors, micro-chemical environments, and even matrix degradations in such a complex synthetic 3D micro-mechano-environment need to be further regulated to create true 3D in vitro tissue models, the technique demonstrates here the guiding of the heterogeneous 3D tissue reconstruction simply by spatial registration of micromechanical cues. In addition, the observed spatial cellular organization may be also ascribed to spatial variation of free methacrylate groups and their cytotoxic effects[36]. To evaluate the potential cytotoxic effects, we examined the cell viability by fluorescence-based live/dead staining using cell-permeable Calcein-AM in combination with a plasma membrane-impermeable DNA-binding dye propidium iodide[37]. We found most of cells are alive (>90%) after encapsulating cells inside soft PEGDMA hydrogels and culturing for 1 day, which is in agreement with previous findings[23,38]. However, a more thorough examination of the cytotoxicity is necessary.

## Discussion

In this study, we report an oxygen inhibition assisted 3D stereolithography technique that creates arbitrary micro-mechano-environments with orthogonally defined stiffness and geometries. This was achieved by facile patterning of the crosslink density, and in particular assisted by purposely introducing oxygen inhibition in photo-curable hydrogels. The programmed micro-mechano-environments allow tunable stiffness across an order of magnitude within the physiologically relevant range. The method also demonstrates the controlled 3D cellular organization through preferential cell attachment and directed cell migration, opening a new avenue towards 3D in vitro tissue fabrication.

## Methods

**Hydrogel precursor solution preparation**. The hydrogel precursor solution was prepared by dissolving 0.15% (w/v) TINUVIN 234 UV absorber (Sigma-Aldrich, St. Louis, MO) into 80% (w/v) poly (ethylene glycol) dimethacrylate (PEGDMA, MW 750; Sigma-Aldrich) at 37 °C for 1 h, and subsequently add 20% (w/v) deionized water for 1 h and 0.2% (w/v) lithium phenyl-2,4,6-trimethylbenzoyl-phosphinate (LAP, TCI America, Portland, OR) photoinitiator for 2 h.

**Digital projection stereolithographic 3D printing**. Three-dimensional structures were modeled and sliced into multiple cross-sectional gray-scale images with even thickness using CAD software. The prepared hydrogel precursor solution was sandwiched between an impermeable PDMS-coated glass slide and 3-(trimethoxysilyl)propyl methacrylate-treated glass cover, i.e. build stage as shown in Fig. 1a. The multiple gray-scale images were sequentially projected into the hydrogel precursor solution through a dynamic micro-mirror device (DMD). Each layer was cured by projected UV light and then pulled out of the image plane by a motorized stage. Thereafter, the following layers of structure were sequentially cured and a 3D structure could be formed. The spatial stiffness control is achieved by varying local gray-scale value of the cross-sectional image. The projection of gray-scale images and the motorized stage were computer controlled.

**Characterization of 3D-printed hydrogel structures**. Young's modulus (stiffness) was measured by an atomic force microscope (AFM, 5420 Scanning Probe Microscope, Agilent Technologies). Briefly, force spectroscopy (force-volume mapping) was typically conducted on a square grid (32 by 32) over a 100 μm × 100 μm area. The AFM probe was a Novascan silicon (Si) cantilever with a polystyrene particle (diameter 10 μm) as the tip and has a spring constant of 0.09 N/m. The Hertz contact model was used to derive the Young's modulus by assuming a

Poisson's ratio of the hydrogel $v = 0.5$. All AFM measurements were performed under a phosphate-buffered saline solution.

The feature height, i.e., curing thickness of printed hydrogel structures (hydrated) was measured by profilometer (Dektak 3030, Veeco, Santa Barbara, CA). The morphology of printed hydrogel structures was observed by a scanning electron microscope (SEM, SU 3500, Hitachi). After fabrication, structures were soaked into 100% ethanol solution for 12 h to remove unpolymerized resin and then air dried for half an hour in a fume hood. Dried structures were prepared for SEM imaging with a thin layer of gold.

**Cell culture**. Primary bovine pulmonary arterial smooth muscle cells (bPASMCs; gift from Dr. Kurt Stenmark Lab) were cultured in DMEM (15-018-CV, Corning) supplemented with 10% bovine calf serum (BCS; 100–506, GemCell), 4 mM L-glutamine, 100 IU/mL penicillin, 100 μg/mL streptomycin, and 1% non-essential amino acid in an incubator at 37 °C and 5% CO$_2$. Cells were used at passages of 3–5 for all the experiments.

Prior to cell seeding, the surface of 3D-printed hydrogel structures were activated with sulfo-SANPAH and subsequently functionalized with fibronectin. bPASMCs were seeded at a density of $1 \times 10^4$ (for line structures) or $2.5 \times 10^4$ (for tube structures) cells cm$^{-2}$ in serum-free media, rinsed with PBS after 2 h incubation at 37 °C, and cultured for 1 or 3 days in a growth media with 10% BCS (refreshed on day 2). Cells were fixed in 4% formalin for 15 min, permeabilized with 0.1% Triton X-100 for 15 min, stained for F-actin (FITC-phalloidin, Life Technologies) and nuclei (DAPI, Life Technologies) for 45 min, and imaged by a Yokogawa CSU-X1 spinning disk confocal mounted on a Nikon Ti-E inverted microscope (Nikon Instruments). Cell counts per mm$^2$, cell aspect ratio, i.e., major axis/minor axis of the cells, and cell alignment angle, i.e., the angle between major axis of cells and axis of the line pattern, were measured by ImageJ (10× magnification).

**Statistical analysis**. Statistical differences between compared groups were determined using unpaired $t$-tests using Excel software with a $p$-value <0.05 indicating significance. Samples size is indicated within corresponding figure legends. All data are presented as a mean ± standard deviation (s.d.).

## Data availability

Data supporting the findings of this study are available within the article and its supplementary information files and from the corresponding authors upon request.

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

## Acknowledgements

This work was supported by Soft Materials Research Center under National Science Foundation (MRSEC DMR-1420736 to X. Yin), and the National Institutes of Health (NHLBI R01HL119371 to W. Tan). We would also like to thank BioFrontiers Advanced Light Microscopy Core and Dr. Joe Dragavon for their excellent microscopy and imaging support. Primary bPASMCs were a gift from Dr. Kurt Stenmark (Cardiovascular Pulmonary Research Laboratories, University of Colorado Anschutz Medical Campus).

## Author contributions

H.Y. and Y.D. contributed equally to this work. H.Y., Y.D., W.T., and X.Y. conceived the ideas and designed the experiments. H.Y. and Y.D. conducted the experiments and analyzed the data. Y.Z. designed and helped with AFM measurements. H.Y., Y.D., W.T., and X.Y. interpreted the data and wrote the manuscript.

## Additional information

**Competing interests:** The authors declare no competing interests.

