## [Peer Review File · Nature Communications]

Reviewers' comments:

Reviewer #1 (Remarks to the Author):

This paper presents a method of 3D printing hydrogels that uses oxygen inhibition, not to specifically create a deadzone, as with CLIP, but to inhibit polymerization with the specific purpose of reducing the crosslink density of the PEG based hydrogel. The purpose of controlling the crosslink density is to control the stiffness and thereby control the potential for cell growth and migration as a function of surface stiffness.

The structures and figures in the paper are interesting. It is well known in the literature that the stiffness of a hydrogel can impact cell growth. It is also well known that the stiffness of a hydrogel can be controlled through control of the crosslink density of the hydrogel. It is also well known that the dosage, which can be controlled by light intensity and/or exposure time, will impact the degree of polymerization and therefore the crosslink density of a multifunctional oligomer resin system.

The following article appears to use similar methods regarding spatial control of dosage to control crosslink density and mechanical properties of their parts:

"Production of Materials with Spatially-Controlled Cross-Link Density via Vat Photopolymerization" Gregory I. Peterson, Johanna J. Schwartz, Di Zhang, Benjamin M. Weiss, Mark A. Ganter, Duane W. Storti, and Andrew J. Boydston. ACS Applied Materials & Interfaces 2016 8 (42), 29037-29043, DOI: 10.1021/acsami.6b09768

The following reviews the issue of substrate modulus and tissue growth:

"Tissue Cells Feel and Respond to the Stiffness of Their Substrate" Dennis E. Discher, Paul Janmey, Yu-li Wang, et al. Science 18 Nov 2005: Vol. 310, Issue 5751, pp. 1139-1143 DOI: 10.1126/science.1116995

The statement that "Geometry and stiffness are inherently intertwined in fabrication..." is a very broad statement that must include a range of assumptions regarding both the geometry and the resin formulations that should be explained in the text.

The reduction in double bond conversion to lower the crosslink density and corresponding stiffness of the fabricated material is likely to lead to significant cytotoxicity due to the presence of relatively high concentrations of methacrylate groups in the formed hydrogel. While it is well known that cells show higher growth potential on stiffer substrates, another interpretation of the data provided in this paper is that localized cytotoxicity of the lower crosslinked and softer areas of the material inhibited cell growth in those regions.

Spatial control of physical properties in stereolithography type 3D printing is a highly desirable outcome.

Reviewer #2 (Remarks to the Author):

This paper describes a new layer by layer approach to 3D fabrication, where the authors claim to achieve orthogonal programming of stiffness and topography. Interesting 3D structures are created and analyzed by AFM and the cell experiments are fairly convincing.

However, the fundamental premise that geometry and stiffness are intrinsically linked is incorrect. When producing structures by photochemical methods, if one avoids optical saturation, then size and crosslink density (and corresponding stiffness) will increase with laser fluence until all reactive sites have been utilized. Thus, further exposure neither increases the crosslink density nor the size. This has been demonstrated in the multiphoton excited stereolithography literature, but the same concept would apply here.

The paper is pitched as demonstrating orthogonal programming of stiffness and geometry, but I think this is a little misleading as the effect the authors demonstrate is achieving varying stiffness at the same topography (which is potentially powerful itself). True orthogonal programming would vary both parameters in the same structure.

For further validation, I strongly suggest the use of Flory-Rehner analysis as measure of crosslink density. This can also be done locally on the same structure.

In sum, the paper presents a novel fabrication scheme and the cell results are nice. However, the same results can be achieved by other methods already demonstrated by others. Moreover, the claims are a little oversold, as, in my opinion, these have not been demonstrated by the current data.

Reviewer #3 (Remarks to the Author):

This manuscript reports a technique for 3D microstructures with orthogonal programming of crosslinking density. 3D bioprinting is rapidly growing research field, and I agree that authors' work has shown quite interesting results which can elevate 3D bioprinting technique. Stiffness control of 3D printed objects has been investigated by choosing different prepolymers but orthogonal programming of mechanical property of objects has never been reported. Nevertheless, authors' new methodology needs to be investigated further to specified its potential applications. By reflecting the comments below, I believe that we can further refine the potential scope of the technique. In addition, there are some minor issues in manuscript style which should be revised for quality of paper.

1. In free radical polymerization including photopolymerization, some active radicals remain in polymer matrix when the polymerization is stopped during the process. As a result, there is a chance that crosslinking yield is keep increasing as time goes by; it is called as autoacceleration. Is stiffness, or crosslinking yield, maintained for reasonable timescale?
2. In oxygen inhibited photopolymerization, rate of photolysis of photoinitiator and rate of chain termination by oxygen are competitive. Therefore, both intensity of UV light and total dose of UV light should be considered simultaneously. In this manuscript, authors mentioned UV dosage which refers UV intensity. However, there is no information for total UV dose or UV exposure time for polymerization. Can authors provide a data for crosslinking yield in respect of total UV dose?
3. I'm curious about the resolution of authors' 3D stereolithography technique. How about the smallest feature size of hydrogel structures? I think that stiffness control can be problematic as the size of structure decrease since diffusion of oxygen dissolved in the prepolymer also effect on polymerization. There is an article for controlling the shape of hydrogel microstructures using oxygen inhibition and diffusion of oxygen in prepolymer which proves that oxygen dissolved in prepolymer is matter (T.S. Shim et al., Nature Communications, 6, 6584 (2015)).
4. To emphasize authors' technique, it will be great to demonstrate 3D micropatterns with different stiffness. In the same manner, 3D artificial cellular matrix having different stiffness can be a good demonstration. It has been demonstrated by many researchers using 3D bioprinting technique (A. Munaz et al., Journal of Science: Advanced Materials and Devices, 1, 1 (2016))
5. Authors cited a several newest 3d printing related papers pretty well. I found another one discussing about 3d printed hydrogel objects with variable mechanical stiffness. It does not reduce the novelty of authors' work since they controlled the stiffness by composition of prepolymers but it might be helpful to make readers follow the trend of related technology better (J. Odent et al.,

Advanced Functional Materials, 27, 1701807 (2017)).

6. Some figure captions are duplicated in main text which is not necessary.
7. (minor) Scale bar of inset image in Figure 1a is missing.
8. (minor) line 55, page 2, "Poly(ethylene glycol) dimathacrylate" is typo.

Responses to Reviewer #1

We thank the reviewer for her/his insightful report. We are delighted that the reviewer found “*the structure and figures in the paper are interesting*” and “*spatial control of physical properties in stereolithography type 3D printing is a highly desirable outcome*”. Here we address the comments and technical questions raised by the reviewer with new experimental results and analysis.

COMMENT: *The following article appears to use similar methods regarding spatial control of dosage to control crosslink density and mechanical properties of their parts: "Production of Materials with Spatially-Controlled Cross-Link Density via Vat Photopolymerization" Gregory I. Peterson, Johanna J. Schwartz, Di Zhang, Benjamin M. Weiss, Mark A. Ganter, Duane W. Storti, and Andrew J. Boydston. ACS Applied Materials & Interfaces 2016 8 (42), 29037-29043, DOI: 10.1021/acsami.6b09768.*

Response: We thank the reviewer for bringing the reference to our attention. Both of our works were actually motivated by the exact same observation/limitation in hydrogel stereolithography, *i.e.* the exposure dosage not only affects the stiffness of resultant hydrogel structures but also the dimensions of printed structures. As shown in Fig. R1 below, the printed dimensions are different at different dosages when all other conditions were kept the same (similar observation was reported in Fig. 5 in the aforementioned reference at a much worse spatial resolution).

Figure 5. Difference in *x*- and *y*-axes diameter dimensions of samples printed at lower light intensities in comparison to the specimen printed at 200 klx (see SI for details). The R^2 for the linear regression line is 0.9777.

Figure R1. *The feature sizes of printed structures were sensitive to light dosage. Right image was adapted from G. I. Peterson, et. al. (2016).*

The printed geometry and mechanical property are therefore intrinsically related in the process of photopolymerization of PEG. For large structures (in millimeter scale) demonstrated by G. I. Peterson, *et. al.*, one could adjust the exposure dimensions to achieve targeted structure sizes based on the above pre-calibration curve (Fig. R1). The approach is unfortunately limited in printing 3D structures with arbitrary defined spatially varying mechanical properties, particularly at a higher spatial resolution because such a calibration is highly geometry specific.

We instead addressed this fundamental problem in stereolithography by leveraging the oxygen-inhibition to decouple the stiffness and geometry during the printing process. We have now cited the aforementioned literature as reference 9 and provide a brief discussion in the revised manuscript at Page 1, Paragraph 1:

“Before optical saturation, geometry and stiffness are, however, inherently intertwined in three-dimensional (3D) photopolymerization because of the different extents of light scattering and outgrowth of photopolymerization at different levels of exposures⁹, making it fundamentally challenging to prepare complex 3D structures with independently defined heterogeneous micro-mechano-environments and geometries¹⁰⁻¹⁴.”

COMMENT: *The following reviews the issue of substrate modulus and tissue growth: "Tissue Cells Feel and Respond to the Stiffness of Their Substrate" Dennis E. Discher, Paul Janmey, Yuli Wang, et al. Science 18 Nov 2005: Vol. 310, Issue 5751, pp. 1139-1143 DOI: 10.1126/science.1116995*

Response: We thank the referee for bringing this important review article to our attention. We have now added the citation and referred to it in the revised manuscript at Page 5, Paragraph 3 as reference 33.

COMMENT: *The statement that "Geometry and stiffness are inherently intertwined in fabrication..." is a very broad statement that must include a range of assumptions regarding both the geometry and the resin formulations that should be explained in the text.*

Response: The statement is made based on our observation that the exposure dosage not only determines stiffness of resultant hydrogel structures but also the dimensions of printed structures due to the different extents of light scattering and outgrowth at different levels of exposure dosages. As shown in Fig. R1, the printed dimensions are different at different dosages (similar

observation was reported by *G. I. Peterson, et. al. 2016*). The geometry and stiffness are inherently related in photopolymerization. However, we do agree with the reviewer that the statement is based on the observation of our specific geometry and resin formulations. We have revised the manuscript at Page 1, Paragraph 1:

“... geometry and stiffness are, however, inherently intertwined in three-dimensional (3D) photopolymerization because of the different extents of light scattering and outgrowth of photopolymerization at different levels of exposures...”

In this work, we leveraged the conventionally adverse effect of oxygen-inhibition on free radical polymerization to control the cured dimensions and made it weakly sensitive to light dosage. As shown in Figure R2 below, the numerical study reveals that the thickness of the inhibition layer is nearly constant regardless of UV exposure dosage especially when dosage is high (Fig. R2a). In contrast, the double bond conversion rate increases with exposure dosage but keeps constant through the cured depth (Fig. R2b). We, therefore, were able to decouple the two factors and achieve the orthogonal control of stiffness and geometry: exposure pattern processes the geometry while gray scale intensity controls the stiffness.

Figure R2. a) Depth profile of double bond conversion rate under different UV exposure dosage. b) Curves showing double bond conversion as functions of depth for various UV exposure dosages tested in this study, i.e. 38, 44, 51, 59, 72, 84 mJ/cm².

In addition, we have conducted further experiments to demonstrate the capability of print 3D hydrogel structures with tailored spatial varying mechanical properties. As shown in Fig. R3

below, the cartoon model was printed with a stiff body while a soft heart inside (was shown in the inset of Fig. 1 in the original manuscript), respectively with a dosage of 72 mJ/cm² and 44 mJ/cm². While SEM only reveals tomography, the optical contrast in dark-field image clearly reflects the stiffness differences (Fig. R3c). We have now provided the new data and related discussions in the revised manuscript at Page 5, Paragraph 2, as shown below:

“We further demonstrate the orthogonal control of geometry and stiffness in a complex 3D structure, a cartoon model [redacted] with a stiff body but a soft heart inside (Fig. 3d-f). SEM image reveals well-defined tomography (Fig. 3e) while the apparent optical contrast in dark-field image reflects the differences between the stiff body (bright) and soft heart (dark) (Fig. 3f).”

[redacted]

Figure R3. Example to show the inherently intertwined stiffness and geometry can be decoupled and a cartoon model was printed with a stiff body while a soft heart inside. Scale bars: 500 μm .

Lastly, we further measured the crosslink density of the polymerized structures by performing swelling experiments and Flory-Rehner theory, which once again confirms the dependences of stiffness on UV exposure dosage. As shown in Fig. R4 below, the effective crosslink density increased with UV exposure dosage, which is in consistent with the stiffness measured by AFM (Fig. 2). The data has now been added into the supplementary information as supplementary Fig. 1 and relevant discussion the main text at Page 4, Paragraph 3, as shown below:

“Further examination of the crosslink density in polymerized PEGDMA hydrogels by swelling ratio measurement and Flory-Rehner theory demonstrates the increasing effective crosslink

density with increasing UV dosages (Supplementary Fig. 1)²⁹, which is in consistent with the stiffness.”

Figure R4. The effective crosslink density was measured by swelling experiment and Flory-Rehner calculation under different UV exposure dosage.

COMMENT: The reduction in double bond conversion to lower the crosslink density and corresponding stiffness of the fabricated material is likely to lead to significant cytotoxicity due to the presence of relatively high concentrations of methacrylate groups in the formed hydrogel. While it is well known that cells show higher growth potential on stiffer substrates, another interpretation of the data provided in this paper is that localized cytotoxicity of the lower crosslinked and softer areas of the material inhibited cell growth in those regions.

Response: We thank the reviewer for bringing up a very important point of cytotoxicity. We did not discuss the cytotoxicity in the original manuscript because this is a ubiquitous but complex and materials-specific issue. Systematic research needs to be conducted for a thorough understanding of how varying crosslink density and stiffness affect cytotoxicity of fabricated hydrogel structures. We actually have noticed the issue and here we presented the preliminary study of cell viability evaluation by encapsulating cells inside the 3D soft PEGDMA gels (the

same UV exposure dosage of 44 mJ/cm² as producing the patterned stiffness). As shown in the Fig. R5 below, most cells are alive (green channel) after culturing for 1 day.

Figure R5. Cell viability evaluation by encapsulating cells inside the soft PEGDMA gels and culturing for 1 day. Green: live cells. Red: dead cells.

This data is consistent with previous findings that high cell viability (> 90%) was observed when the cells were cultured with PEGDMA gels [Cui *et al.* (2012), Fairbanks *et al.* (2009)]^{R1,R2}. Therefore, we believe the spatial cellular organization is majorly ascribed to patterned stiffness instead of localized cytotoxicity. However, we do agree more systematic investigation should be performed in answering this important question. In this regard, we added the following in the revised manuscript at Page 7, Paragraph 1:

“In addition, the observed spatial cellular organization may be also ascribed to spatial variation of free methacrylate groups and their cytotoxic effects³⁶. To evaluate the potential cytotoxic effects, we examined the cell viability by fluorescence-based live/dead staining using cell-permeable Calcein-AM in combination with a plasma membrane-impermeable DNA-binding dye propidium iodide³⁷. We found most of cells are alive (> 90%) after encapsulating cells inside soft PEGDMA hydrogels and culturing for 1 day, which is in agreement with previous findings^{24,38}. However, a more thorough examination of the cytotoxicity is necessary.”

References

[R1] Cui, X., Breitenkamp, K., Finn, M. G., Lotz, M., & D'Lima, D. D. (2012). Direct human cartilage repair using three-dimensional bioprinting technology. *Tissue Engineering Part A*, 18(11-12), 1304-1312.

[R2] Fairbanks, B. D., Schwartz, M. P., Bowman, C. N., & Anseth, K. S. (2009). Photoinitiated polymerization of PEG-diacrylate with lithium phenyl-2, 4, 6-trimethylbenzoylphosphinate: polymerization rate and cytocompatibility. *Biomaterials*, 30(35), 6702-6707.

Responses to Reviewer #2:

We thank the reviewer for her/his constructive report and her/his encouraging remarks that our work presents “*a novel fabrication scheme and the cell results are nice*” and creates “*interesting 3D structures*” as well as “*AFM and cell experiments are fairly convincing*”. Here we address the comments and the technical questions from the reviewer with the new experimental evidences and analysis.

COMMENT: *The fundamental premise that geometry and stiffness are intrinsically linked is incorrect. When producing structures by photochemical methods, if one avoids optical saturation, then size and crosslink density (and corresponding stiffness) will increase with laser fluence until all reactive sites have been utilized. Thus, further exposure neither increases the crosslink density nor the size. This has been demonstrated in the multiphoton excited stereolithography literature, but the same concept would apply here.*

Response: When “*all reactive sites have been utilized*”, we agree with the reviewer that the stiffness will be saturated, and so does the size and crosslink density. The exposure dosage in a multi-photon process is usually optimized to achieve or avoid the saturation regime, which is typically determined by applications [e.g., Pitts *et. al.* (2000) avoids saturation for high resolution multi-photon free-form fabrication^{R1}]. In our work, we also avoid saturation to photopolymerize structures with varying crosslink densities.

In this regime, the dimension and stiffness of a cured structure are intrinsically linked. As shown in Fig. R1, the cured structure dimensions increase with exposure dosages, which becomes the basic bottleneck in stereolithography. Similar observations have been widely reported, e.g. Peterson *et. al.* (2016)^{R2}. We instead addressed this fundamental problem in stereolithography by leveraging the oxygen-inhibition to decouple the stiffness and geometry during the printing process. To clarify and provide a more specific discussion, we have now added more explanation in the revised manuscript at Page 1, Paragraph 1, as shown below:

“Before optical saturation, geometry and stiffness are, however, inherently intertwined in three-dimensional (3D) photopolymerization because of the different extents of light scattering and outgrowth of photopolymerization at different levels of exposures⁹, making it

fundamentally challenging to prepare complex 3D structures with independently defined heterogeneous micro-mechano-environments and geometries¹⁰⁻¹⁴.”

Figure R1. *The feature sizes of printed structures were sensitive to light dosage.*

COMMENT: *The paper is pitched as demonstrating orthogonal programming of stiffness and geometry, but I think this is a little misleading as the effect the authors demonstrate is achieving varying stiffness at the same topography (which is potentially powerful itself). True orthogonal programming would vary both parameters in the same structure.*

For further validation, I strongly suggest the use of Flory-Rehner analysis as measure of crosslink density. This can also be done locally on the same structure.

Response: We thank the reviewer for sharing the same opinion that achieving varying stiffness at the same topography is potentially a powerful technique itself. Our approach does provide the capability. As shown in Fig. R2 below, we show a printed cartoon model with a stiff body but a soft heart (the cartoon was actually shown in the inset of Fig. 1 in the original manuscript). The stiff and the soft parts were printed at 72 mJ/cm² and 44 mJ/cm², respectively. While the SEM images (Fig. R2b) only reveals tomography, the optical contrast in dark-field image clearly reflects the stiffness differences (Fig. R2c). We have now provided the new data and related discussions in the revised manuscript at Page 5, Paragraph 2, as shown below:

“We further demonstrate the orthogonal control of geometry and stiffness in a complex 3D structure, a cartoon model [redacted] with a stiff body but a soft heart inside (Fig. 3d-f). SEM image reveals well-defined tomography (Fig. 3e) while the apparent optical contrast in dark-field image reflects the differences between the stiff body (bright) and soft heart (dark) (Fig. 3f).”

We have now also included these figures (Fig. R2a-c) in Fig. 3 of the revised manuscript.

[redacted]

Figure R2. *Example to show the inherently intertwined stiffness and geometry can be decoupled and a cartoon model was printed with a stiff body while a soft heart inside. Scale bars: 500 μm .*

Following the suggestion, we further measured the crosslink density of PEGDMA hydrogels under different UV exposure dosage by performing swelling experiments using Flory-Rehner theory. As shown in Fig. R3 below, the effective crosslink density increased with UV exposure dosage, which is in consistent with the stiffness measured by AFM (Fig. 2). The data has now been provided in the supplementary information as Supplementary Fig. 1. We have also revised the manuscript at Page 4, Paragraph 3, as shown below:

“Further examination of the crosslink density in polymerized PEGDMA hydrogels by swelling ratio measurement and Flory-Rehner theory demonstrates the increasing effective crosslink density with increasing UV dosages (Supplementary Fig. 1)²⁹, which is in consistent with the stiffness.”

Figure R3. The effective crosslink density was measured by swelling experiment and Flory-Rehner calculation under different UV exposure dosage.

References

[R1] Pitts, J. D., Campagnola, P. J., Epling, G. A., and Goodman, S. L. (2000). Submicron multiphoton free-form fabrication of proteins and polymers: studies of reaction efficiencies and applications in sustained release. *Macromolecules*, 33(5), 1514-1523.

[R2] Peterson, G. I., Schwartz, J. J., Zhang, D., Weiss, B. M., Ganter, M. A., Storti, D. W., and Boydston, A. J. (2016). Production of materials with spatially-controlled cross-link density via vat photopolymerization. *ACS applied materials & interfaces*, 8(42), 29037-29043.

Responses to Reviewer #3:

We thank the reviewer for her/his insightful report to our work. We are delighted with the referee's positive remarks that our work shows "*quite interesting results which can elevate 3D printing technique*" and "*orthogonal programming of mechanical properties of objects have never been reported*". Here we address the comments and the technical questions from the reviewer with the new experimental evidences and analysis.

COMMENT: *In free radical polymerization including photopolymerization, some active radicals remain in polymer matrix when the polymerization is stopped during the process. As a result, there is a chance that crosslinking yield is keep increasing as time goes by; it is called as autoacceleration. Is stiffness, or crosslinking yield, maintained for reasonable timescale?*

Response: The reviewer brought up an important point of autoacceleration, which is common in photopolymerization of multifunctional monomers due to the hindered mobility of reactive chain ends. The autoacceleration usually lasts for several seconds to minutes before the polymerization rate falls off substantially because the propagation reactions become diffusion-controlled with further increased cross-link density [e.g., Anseth et al. (1994); Lovell et al. (1999); Yu et al. (2001)]^{R1-R3}. In our study, all the stiffness measurements were performed after overnight (> 12 h) swelling of 3D printed structure in this study. Therefore, we believe the crosslinking yield has been stabilized before our measurement of stiffness. To address the reviewer's concern, we have now added the relevant discussion in the revised manuscript at Page 4, Paragraph 2, as shown below:

"All stiffness measurements were performed after overnight (> 12 h) swelling post-printing to ensure that the polymerization of remaining free radicals as well as hydrogel swelling were complete."

COMMENT: *In oxygen inhibited photopolymerization, rate of photolysis of photoinitiator and rate of chain termination by oxygen are competitive. Therefore, both intensity of UV light and total dose of UV light should be considered simultaneously. In this manuscript, authors*

mentioned UV dosage which refers UV intensity. However, there is no information for total UV dose or UV exposure time for polymerization. Can authors provide a data for crosslinking yield in respect of total UV dose?

Response: All the data we showed in this manuscript is total UV dosage (mJ/cm^2), *i.e.* UV intensity (mW/cm^2) \times exposure time (s). The UV intensity and exposure time are both important for control of polymerization. The UV intensity determines the rate of initiation and then the rate of polymerization. Under a certain UV intensity, a longer UV exposure yields a higher crosslink density and then stiffness before the saturation of polymerization reaction. We therefore provided the numerical simulation results of double bond conversion rate, *i.e.* crosslink density as functions of both UV exposure time (left panel of Fig. R1 below) and UV intensity (right panel of Fig. R1 below).

Figure R1. Depth profile of double bond conversion rate under different UV exposure time (left) and intensity (right).

The double bond conversion increases with increasing UV exposure time or UV intensity, resulting in higher crosslink density and then stiffness. It is interesting to note that the thickness of oxygen inhibition layer, *i.e.* the un-crosslinked bottom layer is only weakly sensitive to UV intensity and remains constant with the change of UV exposure time. Because the competitive balance between free radical initiation and oxygen scavenging, once reached at threshold UV exposure time ($\sim 4\text{ s}$ in our study as shown in left panel of Fig. R1), is maintained

throughout the remaining reactions. Similar theoretical modeling and observations on the dynamics were also reported by O'Brien and Bowman et al. (2006)^{R4}.

COMMENT: *I'm curious about the resolution of authors' 3D stereolithography technique. How about the smallest feature size of hydrogel structures? I think that stiffness control can be problematic as the size of structure decrease since diffusion of oxygen dissolved in the prepolymer also effect on polymerization. There is an article for controlling the shape of hydrogel microstructures using oxygen inhibition and diffusion of oxygen in prepolymer which proves that oxygen dissolved in prepolymer is matter (T.S. Shim et al., Nature Communications, 6, 6584 (2015)).*

Response: The reviewer raises an important issue of printing resolution. The designed optical resolution in our system is 2 μm per pixel, limited by the spatial resolution of the dynamic mirror device (DMD). However, the actual resolution achievable in stereolithography is poor than this due to the specific resin used and the presence of oxygen diffusion and inhibition. Moreover, the smallest feature size with well-defined stiffness does vary with the exposure dosages, *i.e.*, the stiffer the structures the smaller features one can fabricate. The reviewer is absolutely right that the oxygen inhibition and diffusion in pre-polymer play a key role, which has also been well explained in the Shim's paper mentioned by the reviewer [Shim et al. (2015)]. To address this, we printed the smallest feature sizes of a well-defined rod structure with different stiffness. As shown in Fig. R2 below, the smallest feature size is sensitive to the variation of stiffness (exposure dosage). A smaller feature size can be achieved with a stiff rod, which is most likely due the oxygen diffusion and inhibition. We have now provided the results in the supplementary information as Supplementary Fig. 2 and revised the manuscript at Page 5, Paragraph 1, as shown below:

“Moreover, oxygen diffusion and inhibition also affect the minimal feature sizes that could be achieved³⁰. A higher dosage exposure not only results in a stiffer structure but also allows printing a smaller feature. We have achieved approximately 20 μm well-defined rod structures at a dosage of 84 mJ/cm^2 (Supplementary Fig. 2).”

Figure R2. The smallest feature size that can be produced under different UV exposure dosage. The 3D printed rods were dried, imaged by SEM, and measured by ImageJ.

COMMENT: To emphasize authors' technique, it will be great to demonstrate 3D micropatterns with different stiffness. In the same manner, 3D artificial cellular matrix having different stiffness can be a good demonstration.

Response: Following the reviewer's suggestion, we have designed and printed an animal model with stiff body while a soft heart inside by mimicking the physiological-relevant organ stiffness (shown in Fig. R3 below). The stiffness variation between stiff body and soft heart was revealed by the strong optical contrast in the dark-field optical image (Fig. R3c). We have now provided the data and relevant discussion in the revised manuscript at Page 5, Paragraph 2, as shown below:

“We further demonstrate the orthogonal control of geometry and stiffness in a complex 3D structure, a cartoon model [redacted] with a stiff body but a soft heart inside (Fig. 3d-f). SEM image reveals well-defined tomography (Fig. 3e) while the apparent optical contrast in dark-field image reflects the differences between the stiff body (bright) and soft heart (dark) (Fig. 3f).”

[redacted]

Figure R3. Example to show the inherently intertwined stiffness and geometry can be decoupled and a cartoon model was printed with a stiff body while a soft heart inside. Scale bars: 500 μm .

COMMENT: Authors cited a several newest 3d printing related papers pretty well. I found another one discussing about 3d printed hydrogel objects with variable mechanical stiffness. It does not reduce the novelty of authors' work since they controlled the stiffness by composition of prepolymers but it might be helpful to make readers follow the trend of related technology better (J. Odent et al., *Advanced Functional Materials*, 27, 1701807(2017)).

Response: We agree with the reviewer that the suggested paper is an interesting work to control the hydrogel stiffness by adjusting the pre-polymer formulation. We have added the citation and referred to it in the revised manuscript as reference 14.

COMMENT: Some figure captions are duplicated in main text which is not necessary.

Response: We have removed the duplicated figure captions from the main text (mainly from Figure 2 and 3).

COMMENT: (minor) Scale bar of inset image in Figure 1a is missing.

Response: We have added the scale bar of inset image in Figure 1a caption.

COMMENT: (minor) line 55, page 2, "Poly(ethylene glycol) dimathacrylate" is typo.

Response: We have corrected the typo. We have also thoroughly proof read the revised manuscript for typo, for which we are grateful.

References

[R1] Anseth, K. S., Wang, C. M., & Bowman, C. N. (1994). Kinetic evidence of reaction diffusion during the polymerization of multi (meth) acrylate monomers. *Macromolecules*, 27(3), 650-655.

[R2] Lovelh, L. G., Newman, S. M., & Bowman, C. N. (1999). The effects of light intensity, temperature, and comonomer composition on the polymerization behavior of dimethacrylate dental resins. *Journal of Dental Research*, 78(8), 1469-1476.

[R3] Yu, Q., Zeng, F., & Zhu, S. (2001). Atom transfer radical polymerization of poly (ethylene glycol) dimethacrylate. *Macromolecules*, 34(6), 1612-1618

[R4] O'Brien, A. K., & Bowman, C. N. (2006). Modeling the effect of oxygen on photopolymerization kinetics. *Macromolecular Theory and Simulations*, 15(2), 176-182

REVIEWERS' COMMENTS:

Reviewer #2 (Remarks to the Author):

The authors have addressed all my original concerns. Based on my reading, they have also done a good job addressing the other critiques as well. I have no remaining concerns.

Reviewer #3 (Remarks to the Author):

Authors reflect reviewers' comments comprehensively in revised manuscript. In situ control of crosslinking density of hydrogel polymer in 3d microstructure is quite interesting. Authors showed technological potentials using both quantitative and qualitative analysis. Therefore, I think it is worth to be published in Nature Communications. For better manuscript, I just want to point out one thing that I already did in previous review.

1. As I commented in previous review, however, demonstration of multi-patterned 3d microstructures will make the manuscript more solid. For example, Figure 4 only show microstructures having two different domains: stiff and soft domains. To emphasize programmability of authors' strategy, microstructures with either more than three domains or gradient domains can be demonstrated.

Responses to Reviewer #3:

We are delighted that the reviewer found “*authors reflect reviewers’ comments comprehensively in revised manuscript*” and “*in situ control of crosslinking density of hydrogel polymer in 3d microstructure is quite interesting*”. We thank the reviewer for her/his recommendation for publication. Here we provide responses to address the following technical comment with our new experiments results.

COMMENT: ... however, demonstration of multi-patterned 3d microstructures will make the manuscript more solid. For example, Figure 4 only show microstructures having two different domains: stiff and soft domains. To emphasize programmability of authors’ strategy, microstructures with either more than three domains or gradient domains can be demonstrated.

Response: Multi-patterned 3D microstructures can be directly fabricated with our current printing technique. Following the reviewer’s suggestion, we have conducted new experiments to demonstrate such a programmability by printing multiple 3D structures with different stiffness simultaneously. Using the structure shown in Fig. R1 below as one example, four rods of identical geometry were printed with spatially varying UV exposure dosages of 44, 51, 59, and 72 mJ/cm², which corresponding to the stiffness of approximately 5, 7, 8, and 11 kPa, respectively. When removed from solution and dried in air, the rods show different degrees of bending as the results of different stiffness. The data has now been provided in the supplementary information as supplementary Fig. 3 and referred in the revised manuscript at Page 6, Paragraph 1, as shown below:

“In addition, to show our capability of programming multiple stiffness domains simultaneously, four rods with similar geometries were printed with spatially varied UV exposure dosages and therefore different stiffness (Supplementary Fig. 3). When released from solution and dried in air, different degrees of bending were observed as the results of spatially varying stiffness.”

Stiffness increases

Figure R1. A 3D structure with four rods sitting on a disk which were printed with the same geometry but four spatially varied UV exposure dosages of 44, 51, 59, and 72 mJ/cm². When removed from solution and dried in air, the printed four rods show different degrees of bending due to different stiffness. Left panel is schematic illustration and right panel is SEM image of printed structure after air drying. Scale bar is 200 μm.